# High Temperature Alters Leaf Lipid Membrane Composition Associated with Photochemistry of *PSII* and Membrane Thermostability in Rice Seedlings

**DOI:** 10.3390/plants11111454

**Published:** 2022-05-30

**Authors:** Paphitchaya Prasertthai, Warunya Paethaisong, Piyada Theerakulpisut, Anoma Dongsansuk

**Affiliations:** 1Department of Agronomy, Faculty of Agriculture, Khon Kaen University, Khon Kaen 40002, Thailand; papichaya_p@kkumail.com; 2Salt Tolerant Rice Research Group, Khon Kaen University, Khon Kaen 40002, Thailand; warunya26@gmail.com (W.P.); piythe@kku.ac.th (P.T.); 3Department of Biology, Faculty of Science, Khon Kaen University, Khon Kaen 40002, Thailand

**Keywords:** fatty acid, heat stress, rice, photosynthesis, membrane stability

## Abstract

Rice cultivated in the tropics is exposed to high temperature (HT) stress which threatens its growth and survival. This study aimed at characterizing the HT response in terms of *PSII* efficiency and membrane stability, and to identify leaf fatty acid changes that may be associated with HT tolerance or sensitivity of rice genotypes. Twenty-eight-day-old seedlings of two Thai rice cultivars (CN1 and KDML105), a standard heat tolerance (N22), and a heat sensitive (IR64) rice genotype were treated at 42 °C for 7 days. Under HT, N22 showed the highest heat tolerance displaying the lowest increase in electrolyte leakage (EL), no increments in malondialdehyde (MDA) and stable maximum quantum yield of *PSII* efficiency (*F_v_/F_m_*). Compared to KDML105 and IR64, CN1 was more tolerant of HT, showing a lower increase in EL and MDA, and less reduction in *F_v_/F_m_*. N22 and CN1 showed a higher percentage reduction of unsaturated fatty acids (C18:2 and C18:3), which are the major components of the thylakoid membrane, rendering the optimum thylakoid membrane fluidity and intactness of *PSII* complex. Moreover, they exhibited sharp increases in long-chain fatty acids, particularly C22:1, while the heat sensitive IR64 and KDML105 showed significant reductions. Dramatic increases in long-chain fatty acids may lead to cuticular wax synthesis which provides protective roles for heat tolerance. Thus, the reduction in unsaturated fatty acid composition of the thylakoid membrane and dramatic increases in long-chain fatty acids may lead to high photosynthetic performance and an enhanced synthesis of cuticular wax which further provided additional protective roles for heat tolerance ability in rice.

## 1. Introduction

High temperature (HT) is one of the main limiting factors in crop growth and development. It leads to morphological, physiological and biochemical changes in crops and restricts crop survival [1] and their productivity by negatively affecting the photosynthetic process [2]. It also causes a loss in membrane integrity [3,4], aging in plant cells, inhibition of plant growth and cell death [5,6]. A temperature greater than 40 °C affects many crops by causing thermal injury, inhibiting photosynthesis and changing protein structure and function such as protein unfolding, denaturation, aggregation and inactivation of enzyme reactions [1]. The first and highly sensitive photosynthetic site which is affected by HT is photosystem II (*PSII*) [7]. HT affects *PSII* by significantly decreasing its function. Several aspects of HT effects on *PSII* have been reported, including the disassociation of the *PSII* peripheral antenna complex from the core complex, inactivation and disassociation of oxygen-evolving *PSII* complexes [8,9], and inhibition of electron transport from Q_A_ to Q_B_ [9,10]. The photosynthetic carbon fixation is disrupted by temperatures above optimum for plant function. This commonly coincides with chlorophyll degradation (caused by reactive oxygen species; ROS) resulting in rapid leaf senescence [11,12].

The cellular membrane is the most sensitive site affected by environmental stimulation [13,14], including HT [9]. HT has been reported to be associated with disrupting cell membranes, breaking down cell compartmentalization [15], leading to solute leakage and affecting cellular functions that cause physiological injuries and leaf senescence [16,17]. The disruption of a cellular membrane by HT is related to the increase in membrane lipid peroxidation, which occurs when double bonds of unsaturated fatty acids are attacked by ROS [18,19].

The fluidity of lipid membrane is important for membrane function that depends on the interaction between lipid and protein compositions. The increase in membrane fluidity resulted from an HT-induced higher degree of lipid unsaturation [20,21]. Thus, to decrease membrane fluidity under HT, lipid unsaturations could be decreased by downregulating the fatty acid desaturase (FAD) gene [22]. Maintaining the optimum composition of unsaturated fatty acids is important in determining cellular membrane fluidity under HT [23]. For heat-tolerant plants under HT, the degree of saturated fatty acids increases in both thylakoid membranes [24] and cellular membranes [25]. This enhances lipid heat stability resulting in thermal tolerance in diverse species [26,27]. Some studies showed that changed lipid saturation was involved in the plant’s ability to tolerate heat. For example, roots of heat-tolerant creeping bentgrass exhibited increased lipid saturation under HT [15]. A heat-tolerant wheat genotype showed higher amounts of saturated lipids and a decrease in the amount of unsaturated lipids [28]. Similarly, *Brassica carinata* cv. Avanza 641 responded to HT by decreasing the level of unsaturation in membrane lipid, especially in unsaturated linolenic acid (18:3) to maintain optimal membrane fluidity [29]. Moreover, soybean [30] and peanut [31] also responded to HT by decreasing lipid unsaturation levels to maintain membrane fluidity, and the magnitudes of reduction were greater for heat-tolerant than for heat-sensitive genotypes. In contrast, some experiments demonstrated that higher degrees of unsaturated fatty acids can maintain membrane fluidity under HT [32]. For example, tall fescue genotypes that exhibited an increased degree of unsaturated fatty acids showed an improvement in heat tolerance [9].

Rice (*Oryza sativa* L.) is an important food crop and a living factor for the world population. Especially in Asia, rice is necessary for household consumption, food products and the national economy [33]. Rice can be cultivated and grows well in a wide geographical and climatic range. Elevated temperature can impose adverse effects on rice at different growth stages (such as germination, seedling, tillering, reproductive and grain filling stages) and induce reductions in rice yield [34]. At the flowering stage, HT reduces spikelet fertility by inhibiting anther dehiscence and hence decreasing pollination. Additionally, at panicle initiation (PI), HT causes spikelet degeneration and damages the development of flower organs. The reduction in spikelet fertility can lead to a severe reduction in rice yield [35]. Several reports in rice and other plants showed that HT adversely affected *PSII* efficiency. A lower maximum quantum yield of *PSII* efficiency (*F_v_/F_m_*) was found in tall fescue grass exposed to 35/30 °C (day/night) for 20 days [36], and in spinach stressed at 50 °C [37]. In rice, significantly reduced *F_v_/F_m_* was reported when plants were subjected to heat stress at 40–55 °C for 30 min at seedling [38], 35 and 40 °C at tillering [39], 45 °C for 30 min at heading [40] and 42 °C for 30 min at dough grain stage [41]. The decline in photosynthetic pigments as affected by HT was found in tall fescue grass [36], in seedlings of rice cv. Riceberry [38], and in rice cvs. IR64 and KDML105 at the dough grain stage [41]. Moreover, HT also stimulated lipid peroxidation, resulting in a decline in cellular membrane thermal tolerance [39]. Previous reports in rice showed that HT led to significantly increased lipid peroxidation and electrolyte leakage [42], in cvs. Dular, IR64 and KDML105 at seedling [43], and in cvs. IR64 and KDML105 at the dough grain stage [41]. HT led to changes in lipid membrane structure and membrane fluidity relating to changes in lipid composition and saturation level [17]. Increased lipid saturation level was reported in creeping bentgrass exposed to high soil temperature (35 °C) [17] or high ambient temperature (35 °C) [15]. In tall fescue turfgrass, increased lipid unsaturation was reported when exposed to HT (40/35 °C, 14/10 h) [9], but in wheat under HT, there was a decline in the degree of lipid unsaturation [28]. Several reports pronounced the alteration of lipid composition upon heat stress such as the increase in degree of saturated fatty acids [15,16,44] or decrease in unsaturated fatty acids [28,44]. Hu et al. [9] proposed that an increase in unsaturated fatty acids enhanced lipid thermostability in a photosynthetic membrane and might contribute to heat tolerance in various species. Therefore, the experimental results inferring the relationship between lipid composition (degree of fatty acid saturation/unsaturation) and heat tolerance in various species are still inconclusive. Thus, this research studied the HT responses of two Thai rice cultivars compared to a standard heat tolerance and a heat-sensitive rice genotype, in terms of physiological parameters and leaf fatty acid composition changes.

## 2. Results

### 2.1. PSII Efficiency after High Temperature Exposure

Rice plants of all four cultivars in the control condition had the maximum quantum yield of PSII efficiency in the dark-adapted state (*F_v_/F_m_*) values equal to or higher than 0.80, indicating the absence of damage to PSII, with CN1 showing the highest value at 0.822 (Figure 1). However, when subjected to HT stress, *F_v_/F_m_* significantly decreased in IR64 (the heat sensitive check), KDML105 and CN1. In contrast, the heat-tolerant genotype (N22) maintained a high *F_v_/F_m_* value (approximately 0.806). Heat stressed CN1 still had relatively high *F_v_/F_m_* that was slightly higher than N22.

### 2.2. Leaf Membrane Stability after High Temperature Exposure

All four rice genotypes under the control condition displayed good membrane stability as indicated by low EL values ranging from 3.24% to 4.16% (Figure 2A). When subjected to HT stress, cellular membranes were damaged, and all rice genotypes showed significantly higher EL. The membranes of the heat-sensitive check, IR64, was most seriously damaged, showing a nine-fold increase in EL, while those of N22 showed the highest integrity with only a 7.15% increase in EL. Of the two Thai cultivars, CN1 maintained greater membrane stability with its EL value similar to that of N22. Loss of membrane stability is partly imposed by peroxidation of membrane lipids, which can be indicated by malondialdehyde content. In the non-stress plants, leaf MDA contents in the four rice genotypes were in the range of 19.19 to 20.63 g gFW^−1^ (Figure 2B), with KDML105 having slightly higher values than others. HT stress resulted in significantly increased MDA contents in CN1, KDML105 and IR64. In contrast, MDA content of N22 was significantly lower than the control. The highest degree of increment in MDA content was found in KDML105 (65.92% increase), followed by IR64 (37.22% increase) and CN1 (21.95% increase).

### 2.3. Alteration of Leaf Total Fatty Acids after High Temperature Exposure

Total fatty acids in seedlings of all four rice cultivars after an exposure to HT sharply reduced by 1.4–2.0-fold compared to the controls (Figure 3). Significant differences in total fatty acids were noted among genotypes. The highest total fatty acid content was found in CN1 for both the non-stress (10.769 µg µL^−1^) and HT-stressed (7.608 µg µL^−1^) plants. The HT-stressed plants of N22, KDML105 and IR 64 had similar contents of total fatty acids, varying from 3.89 to 7.61 µg µL^−1^. The highest percentage reduction in total fatty acids after HT exposure was found in KDML105 at 50.94% and the lowest in CN1 (29.35%).

Total fatty acid was typically composed of total saturated and total unsaturated fatty acid as shown in Figure 4. Total saturated fatty acid contents in seedlings of all rice cultivars were significantly lower than total unsaturated fatty acids. Only N22 and KDML105 showed slightly significant reductions in total saturated fatty acid contents after exposure to HT. The highest and the lowest total saturated fatty acid contents after HT treatment were found in CN1 (1.606 µg µL^−1^) and KDML105 (0.995 µg µL^−1^), respectively. In contrast, unsaturated fatty acid contents in all rice cultivars were markedly reduced compared to the controls. KDML105 showed the sharpest decline in total unsaturated fatty acids (2.6-fold reduction from the control), and the amount under HT was lowest in this cultivar. Conversely, the lowest reduction in total unsaturated fatty acid under HT was found in CN1.

### 2.4. Alteration of Leaf Saturated Fatty Acid Composition after High Temperature Exposure

A total of nine saturated fatty acids were identified from rice seedlings, and the content of each acid under the control and HT treatment are displayed in Table 1. Eight saturated fatty acids, namely lauric acid (C12:0), myristic acid (C14:0), pentadecanoic acid (C15:0), palmitic acid (C16:0), stearic acid (C18:0), arachidic acid (C20:0), behenic acid (C22:0) and lignoceric acid (24:0), were found in seedlings of all four rice cultivars in both control and HT conditions. Only one saturated fatty acid, namely capric acid (C10:0), was found only in Thai rice, CN1 and KDML105. Moreover, capric acid contents in HT-stressed CN1 and KDML105 exhibited significant increments. Palmitic acid (C16:0) was found in the greatest amount under both control and HT conditions. Palmitic acid and stearic acid markedly decreased with HT in all cultivars. Three saturated fatty acids including arachidic acid, behenic acid, and lignoceric acid in N22, CN1 and IR64 were significantly increased after HT treatment, but in KDML105, the contents of these acids declined significantly.

### 2.5. Alteration of Leaf Unsaturated Fatty Acid Composition after High Temperature Exposure

A total of nine unsaturated fatty acids were identified, and the contents under the control and HT treatment are shown in Table 2. Six-unsaturated fatty acids including linoleic acid (C18:2), γ-Linolenic acid (C18:3n6), linolenic acid (C18:3), cis-5,8,11,14,17-eicosapentaenoic acid (C20:5), oleic acid (C18:1) and erucic acid (C22:1) were found in seedlings of all four rice cultivars under both conditions. Cis-11,14-eicosadienoic acid (C20:2) which was present in the non-stress seedlings of all cultivars completely disappeared from all HT-stressed seedlings. In the control seedlings, this acid was found in slightly greater amounts in the Thai rice, CN1 and KDML105. In the non-stress plants, palmitoleic acid (C16:1) was present in N22 and IR64 but absent from CN1 and KDML105. Upon HT stress, the content of this acid significantly increased in all cultivars. The content of palmitoleic acid in the HT plants of CN1 and KDML105 was present several folds higher than those in N22 and IR64.

Changes in mono- and polyunsaturated fatty acids are shown in Figure 5. Monounsaturated fatty acids in all rice cultivars were several folds lower than polyunsaturated fatty acids. Upon HT exposure, monounsaturated fatty acids significantly increased in N22 and IR64, but significantly decreased in Thai rice, CN1. Polyunsaturated fatty acids in seedlings of all rice cultivars markedly decreased under HT. Thai rice, CN1 exhibited the highest level of polyunsaturated fatty acid compared to other cultivars in both control and HT.

### 2.6. Alteration of Leaf Ratio of Saturated and Unsaturated Fatty Acids after High Temperature Exposure

The highest Sat:Unsat under both control and HT condition were found in KDML105 (Table 3). Sharp increases in Sat:Unsat after exposure to HT occurred in all rice cultivars in the range between 1.5–1.8-fold compared to the controls. The highest increase in Sat:Unsat was found in N22, and the lowest increase was found in CN1.

## 3. Discussion

High temperature (HT) is harmful to plant growth and survival. It decreases photosynthetic efficiency and yield [28]. HT can induce excessive reactive oxygen species (ROS) generation, resulting in cellular membrane injury, impaired photosynthetic mechanism and oxygen evolving complex associated with *PSII*, and inhibition of protein synthesis [45]. Plants can adapt to HT by changing the morphology and altering metabolism via signal transduction [9]. The chloroplast photosynthetic function is supposed to be the most HT-sensitive cellular function, and *PSII* is one of the most sensitive to HT [9,46]. In this study, *PSII* efficiency was determined by measuring the maximum quantum yield in the dark-adapted stage (*F_v_/F_m_*), which decreased significantly after exposure to HT in CN1, KDML105 and IR64 but remained unchanged in the heat-tolerant genotype, N22 (Figure 1). Heat-induced reduction in *F_v_/F_m_* resulted from an increase in minimum fluorescence in the dark-adapted state (*F*_0_) which indicated the inhibition of electron transport and a reduction in maximum fluorescence in the dark-adapted state (*F_m_*) that is caused by a dissociation of peripheral antenna complex of PSII from its core complex [47]. Similarly, Dongsansuk et al. [40] reported that *F_v_/F_m_* significantly reduced in the heat-sensitive rice PT60 but was stable in Dular after exposure to 45 °C for 30 min. Dular is known to be a heat-tolerant rice variety comparable to N22 [48]. However, Sailaja et al. [49] stated that in response to heat stress, *F_v_/F_m_* in N22 was stable at the reproductive stage but reduced at the vegetative stage. This indicated that the ability to maintain *PSII* efficiency varies with developmental stage. However, Thai rice cv. CN1 showed the highest *F_v_/F_m_* level, although its *F_v_/F_m_* declined after HT exposure (Figure 1).

The cellular membrane is the most sensitive structure to be affected by HT [9,28]. The thylakoid membrane, where the photosynthetic light reaction occurs, is also highly thermal sensitive [50]. HT can induce an increase in membrane fluidity and oxidative burst, leading to lipid peroxidation resulting in malondialdehyde production (MDA) [28] and other changes in lipid membrane structure, which cause membrane injury, electrolyte leakage (EL) and cell death. MDA content and EL are sufficient indicators of membrane thermostability, and they can be considered directly related to the degree of plant damage by HT [9]. In this study, CN1 and N22 exhibited slightly decreased membrane thermostability, as indicated by a small increase in EL and MDA content (Figure 2), but IR64 and aromatic Thai rice KDML105 displayed a sharp increase in EL and MDA content under HT (Figure 2). Pansarakham et al. [41] previously reported that MDA and EL of rice cvs. IR64, KDML105, and PTT1 significantly increased after an exposure to 42 °C for 7 days while those of Dular and N22 remained unchanged. This suggested that HT caused a reduction in membrane thermostability in IR64 and KDML105 and resulted in markedly decreased photosynthetic efficiency of *PSII* (*F_v_/F_m_*) (Figure 1). Thylakoid membrane is the site where light harvesting pigments, *PSII*, *PSI* and electron carrier proteins are embedded. Thus, the thylakoid membrane-increased fluidity due to HT resulted in weakness in the bonding between lipid and electron carrier protein molecules [51]. Consequently, the light reaction taking place in the thylakoid membrane was inhibited, resulting in a reduction in the efficiency of the *PSII* photochemistry as seen in IR64 and KDML105 (Figure 1).

Plant lipids are crucial structural and functional biomolecules found in cell membranes, mitochondria membranes, tonoplasts, chloroplast membranes and other cellular structures. Extracellular lipids are also present in large amounts as components of cutin, suberin and waxes [52]. Diverse plant lipids consist of polyunsaturated fatty acids, mainly linoleic acid (C18:2) and linolenic acid (C18:3), monounsaturated fatty acids, mainly oleic acid (C18:1) and saturated fatty acids, mainly palmitic acid (C16:0) and stearic acid (C18:0) [53]. An insight into the relationship between the observed physiological parameters under HT and alterations in leaf fatty acid content and composition was manifested in this study. Under HT, total fatty acid content decreased in all rice cultivars (Figure 3, Table 1). A comparison between the two Thai rice cultivars revealed that under HT, CN1 showed the lowest reduction compared to the control and contained the highest total fatty acid content, but KDML105 showed the highest reduction and the lowest amount of total fatty acids (Figure 3, Table 1). This suggested that HT significantly changed total leaf fatty acids, and different genotypes were affected to different degrees. The thylakoid membrane is reported to be the most sensitive site severely damaged by heat stress and the degree of damage is related to plant survival [54]. The major types of lipids in the thylakoid membrane are galactolipids (monogalactosyldiacylglycerol, MGDG and digalactosyldiacylglycerol, DGDG), sulfolipids (sulfoquinovosyldiacylglycerol, SQDG), and phospholipids (phosphatidylglycerol, PG). Of all lipids in thylakoids, MGDG and DGDG can reach 52% and 26%, respectively [55]. The ratio of MGDG to DGDG can indicate the photosynthetic membrane stability [56]. The major types of fatty acids that constitute lipids (MGDG, DGDG, SQDG and PG) in the plant thylakoid membrane are palmitic acid (C16:0) [51,57], palmitoleic acid (C16:1) [57], roughanic acid (C16:3) [51,58], oleic acid (C18:1) [57], linoleic acid (C18:2) [59], and linolenic acid (C18:3) [51,57,59]. Unsaturated fatty acids constitute 75% of total fatty acids in the thylakoid membrane, while the saturated fatty acid composition was approximately 25% [60]. The total saturated fatty acid content under HT remained unchanged in CN1 but significantly declined in KDML105 (Figure 4, Table 1). CN1 also contained the highest, and KDML105 the lowest amount of saturated fatty acids under HT. This suggested that, in CN1, lipid saturation level was well maintained under HT when associated with lower lipid peroxidation, a smaller increase in membrane fluidity, and lower EL (Figure 2). A higher lipid saturation level is also associated with greater membrane stability and higher efficiency of *PSII* photochemistry (Figure 1). In contrast, a decreased lipid saturation level in KDML105 led to lower membrane stability, hence more solute leakage (Figure 2), and dissociation of the photosystem and electron transport components, leading to a significant decline in *PSII* efficiency (Figure 1). Total unsaturated fatty acids decreased in all rice cultivars under HT. A lower content of unsaturated fatty acids, namely linolenic acid (C18:2), γ-linolenic acid (C18:3n6) and linolenic acid (C18:3), which are major components of the thylakoid membrane, helped maintain optimal fluidity and prevent the lipid peroxidation process, which predominantly attacks double bonds [17,18]. Moreover, the percentages of reduction in unsaturated fatty acids were greater than those in saturated fatty acids, resulting in an increased ratio between saturated:unsaturated fatty acids (Table 2 and Table 3 and Figure 4). The highest percent increase in the ratio between saturated:unsaturated fatty acids in N22 could be related to its highest thermotolerance, as indicated by the unchanged *F_v_/F_m_* (Figure 1), lowest lipid peroxidation and EL (Figure 2). Among cultivars, N22 showed the highest percentage reductions in major unsaturated fatty acids which constitute the thylakoid membrane (54% and 58% reduction in linolenic acid and linoleic acid, respectively). A decrease in unsaturation or increase in saturation of fatty acids in membrane lipids was known to help maintain appropriate membrane fluidity at high temperature [9]. Similar to our results, HT treatment in *Brassica carinata* resulted in remodelling of leaf lipidome manifested by decreases in unsaturated levels of membrane lipids, especially C18:3, and increases in saturated fatty acids such as C16:0 [29]. These authors proposed that plants can modulate the level of unsaturation of membrane lipids by decreasing the activity of FAD3, FAD7 and FAD8, which catalyse the formation of C18:3, or by increasing the activity of FAD2 and fatty acid synthase (FAS), which catalyse the formation of C18:2 or C16:0, respectively. The time of exposure and the level of temperature elicited different responses in relation to fatty acid changes, which varies depending on plant species. In contrast to this study, Wang et al. [61] reported that for temperate species such as Arabidopsis, heat shock at 38 °C for 6 h did not induce any changes in unsaturated fatty acids but prolonged treatment at warm temperature (28 °C, 7 d), resulting in an enrichment of these acids to achieve an optimum level of membrane fluidity.

Most previous studies focused on the changes in lipid composition of isolated thylakoid membranes under heat stress. Our study analysed fatty acid compositions of the whole leaf; therefore, many fatty acids not previously reported in heat-stress studies were identified and displayed some interesting patterns of responses. Among saturated fatty acids, arachidic acid (C20:0), behenic acid (C22:0) and lignoceric acid (24:0) (Table 1) sharply increased in N22, CN1 and IR64, while these acids significantly reduced in KDML105. In addition, erucic acid (C22:3), a long-chain unsaturated fatty acid, increased in response to HT in N22 and CN1, but significantly reduced in KDML105 and IR64 (Table 2). Long chain fatty acids (C20 to C36) are known to be precursors for cuticular wax synthesis [62]. A significant increase in long-chain fatty acids could be related to enhanced synthesis of cuticular wax and hence thicker cuticles in these cultivars. Thicker cuticles could help reduce excessive transpirational water loss during HT stress and increase light dispersion [63]; therefore, alleviated cellular damage was evidenced by lower EL and higher *F_v_/F_m_*, particularly in N22 and CN1, compared to those of KDML105. Although IR64 showed significant increases in the saturated long-chain fatty acids, its proposed enhanced cuticular wax synthesis might be offset by its weaker antioxidant systems as evidenced by high MDA content, leading to higher ROS-induced cellular damage. Moreover, saturated fatty acid, namely capric acid (C10:0), under HT and control was found in Thai rice, CN1 and KDML105. In particular, KDML105 showed the highest capric acid content, and it increased significantly after exposure to HT (Figure 5, Table 2). Capric acid (C10:0) was not found in N22 and IR64. Capric acid (C10:0) occurs in coconut oil, and it is used as an aromatic chemical in the manufacture of perfumes. This suggested that Thai rice, CN1 and KDML105, especially KDML105 or Thai Jasmine rice, is an aromatic rice and has volatile aromatic compounds that make the aroma and flavour in the rice. Thus, capric acid (C10:0) in Thai rice may involve aroma and flavour, not only 2AP.

## 4. Materials and Methods

### 4.1. Plant Materials and Temperature Condition

Four rice cultivars, namely N22 (heat tolerant rice; [48]), CN1 (Thai rice growing in the central region of Thailand), KDML105 (aromatic Thai rice) and IR64 (heat sensitive rice; [64]) were used as plant materials. Rice seeds were sterilized by soaking in 70% ethanol, followed by 10% Clorox and rinsing three times in tap water. Sterilized seeds were soaked in tap water for 24 h and then germinated on wet filter paper for 7 days. Fifty seedlings were transplanted in a pot size of 12.5 L with 4 pots in each treatment. After they were grown for 28 days in a greenhouse (at Agronomy station, Faculty of Agriculture, Khon Kaen University, Thailand, 16°28′14.6208″ N 102°48′41.3532″ E), seedlings were transferred to a temperature chamber for high temperature (HT) treatment at 42 °C/27 °C (day/night) for 7 days (Table 4 and Table 5) or remained in the greenhouse for control. Climate conditions in the greenhouse were observed during March–December 2019 with average air temperature at 35 °C, relative humidity at 89% and average daily PAR in the range of 1064–1145 µmol m^−2^ s^−1^. After 7 days of temperature treatment, maximum quantum yield of *PSII* efficiency (*F_v_/F_m_*) of seedlings was determined. Leaf samples were collected and immediately investigated for electrolyte leakage (EL). Other leaf samples were kept frozen at −20 °C for the determination of malondialdehyde (MDA) and fatty acid.

### 4.2. Determination of Photosynthetic Efficiency of PSII

Photosynthetic efficiency of *PSII* was determined by measuring maximum quantum yield of *PSII* efficiency in the dark-adapted state (*F_v_/F_m_*; (1)), minimal fluorescence in the dark-adapted state (*F*_0_) and maximal fluorescence in the dark-adapted state (*F_m_*) using a chlorophyll fluorometer (Mini PAM, Walze, Effeltrich, Germany). The maximum quantum yield of *PSII* efficiency (*F_v_∕F_m_*) was calculated as described by Schreiber [65] as follows (1):*F_v_∕F_m_* = (*F_m_* − *F*_0_)∕*F_m_*(1)

### 4.3. Determination of Electrolyte Leakage

The electrolyte leakage (EL) was determined according to Bajji et al. [66] with some modifications. Fresh leaves (0.1 g) were placed in a test tube containing 10 mL of deionized water (DI water) and incubated in a water bath (Stirrer Digital Bath, Clifton, Nickel-Electro Co., Ltd., North Somerset, England) at 32 °C for 2 h. The electrical conductivity of the bathing solution was then measured using an EC meter (Five easy™ plus FEP30, Mettler-Toledo AG, Schwerzenbach, Switzerland) and recorded as EC_1_. After that, the tubes were autoclaved (Tomy Model ES-315, Tomy Seiko Co., Ltd., Tokyo, Japan) at 15 PSI, 121 °C for 15 min. The samples were cooled, and the final electrical conductivity (EC_2_) was recorded. The unit of EL was expressed in % as follows (2):EL (%) = (EC_1_/EC_2_) × 100(2)

### 4.4. Determination of Malondialdehyde Contents

Malondialdehyde (MDA) contents were determined by a modified method of Heath and Packer [67]. Fresh leaves (0.1 g) were placed in 15 mL test tubes and added with 1.4 mL of distilled water. Then, 1.5 mL of 0.5% (*w/v*) thiobarbituric acid (TBA) in 20% (*w/v*) trichloroacetic acid (TCA) was added and mixed. The solution was incubated in a water bath (Stirrer Digital Bath, Clifton, Nickel-Electro Co., Ltd., England) at 100 °C for 25 min. The reaction was stopped by cooling down on ice for 5 min. The absorbance of the reaction mixture was taken at 532 and 600 nm in a UV–Vis spectrophotometer (UV-VIS Spectrophotometer Model i3, Jinan Hanon Instruments Co., Ltd., Shandong, China). The MDA content was calculated according to (3) and expressed as mg gFW^−1^ (3).
MDA contents (mg gFW^−1^) = (OD532 − OD600)/155(3)
where 155 mM^−1^ cm^−1^ is the extinction coefficient of the MDA at a wavelength of 532 nm.

### 4.5. Determination of Fatty Acid

Lipid extraction was modified according to Folch et al. [68]. Leaf samples (50 g) were placed in a beaker and 90 mL of chloroform:methanol (2:2 *V/V*) added. Samples were homogenized and filtered using a vacuum pump filter. The extracted solution was added with 30 mL of chloroform, 30 mL RO water and 50 mL of 0.58% NaCl and was shaken for separating the lipid layer. The lipid layer was placed in a 250 mL Erlenmeyer flask and lipid was extracted one more time. All extracted lipid was added to Na_2_SO_4_ for dehydration and filtered by filter paper no. 1. The solvent was evaporated from lipid extraction until it disappeared completely by Rotary evaporator. Lipid extraction was blown by N_2_ gas until the lipid weight was stable.

Fatty acid contents were determined according to Metcalfe et al. [69]. Lipid-extracted solution was weighed (30 mg) and placed in a test tube. Then, 1.5 mL of 0.5 M Methanolic NaOH was added and boiled in a water bath at 80–90 °C for 2 min. The solution was shaken for breaking the lipid and then cooled at room temperature. Then, 1 mL of internal standard C17 and 2 mL of BF_3_ were added and boiled in the water bath at 80–90 °C for 30 min. Next, 1 mL of Isooctane and 5 mL of 36% NaCl were added and mixed by vortex. Supernatant was sucked and then placed into a test tube. Solvent in the supernatant was evaporated by blowing with N_2_ gas. This supernatant was diluted by 1 mL Isooctane, and fatty acids were analysed by gas chromatography (GC) (Agilent 7890a, Agilent Technologies, Inc., Wilmington, NC, USA)

### 4.6. Statistical Analysis

This research experiment was designed in completely randomized design (CRD) with 4 replications. The difference between control and HT in each rice cultivar was determined by independent-samples *t* test, and comparison of different rice cultivars in control or HT was analysed by analysis of variance (ANOVA) and Duncan’s multiple range tests (DMRT). Statistical analysis was performed using SPSS for Windows version 17.0.

## 5. Conclusions

HT caused differential alterations in fatty acid profiles, especially those comprising the thylakoid membrane in rice cultivars differing in heat tolerance. In the case of N22 and CN1, their leaves slightly decreased the degree of lipid thylakoid membrane composition in the amount of palmitic acid (C16:0), linolenic acid (C18:2), γ-linolenic acid (C18:3n6) and linolenic acid (C18:3) and increased the amount of palmitoleic acid (C16:1). The greater reduction in unsaturated fatty acids under HT, especially in N22, was related to maintenance of optimum thylakoid membrane fluidity and increased membrane stability. Moreover, in N22 and CN1, long-chain fatty acids, which serve as precursors of cuticular wax synthesis, namely arachidic acid (C20:0), behenic acid (C22:0), lignoceric acid (24:0) and erucic acid (C22:1), also increased. These long-chain fatty acids may contribute to increased thickness of the cuticle, which helps prevent excessive water loss and reflects strong light, resulting in lower HT-induced damage. Better maintenance of membrane thermostability and thicker cuticles in heat-tolerant genotypes lead to less electrolyte leakage, lower lipid peroxidation and stable *F_v_/F_m_* compared to the more sensitive genotypes. Thus, this study suggested that there may be some relationships between the lipid composition of the thylakoid membrane and cuticle characteristics in a rice leaf, and their ability to tolerate HT. From this study, changes in leaf fatty acid compositions can potentially be used as indicators for heat tolerance in rice.

## Figures and Tables

**Figure 1 plants-11-01454-f001:**
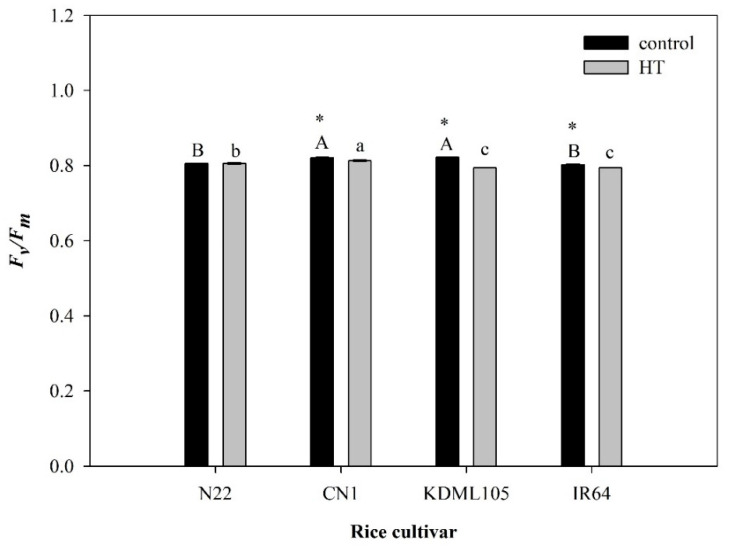
Effect of HT on *F_v_/F_m_* in four rice cultivars (N22, CN1, KDML105 and IR64). Each bar presents mean ± SE (*n* = 3–4). Asterisk (*) indicates significant difference between control and HT in each rice cultivar at *p* ≤ 0.05 by *t* test. Different capital letters indicate significant difference of control group in different rice cultivars at *p* ≤ 0.05 by ANOVA and DMRT, and small letters indicate significant difference of HT in different rice cultivars at *p* ≤ 0.05 by ANOVA and DMRT.

**Figure 2 plants-11-01454-f002:**
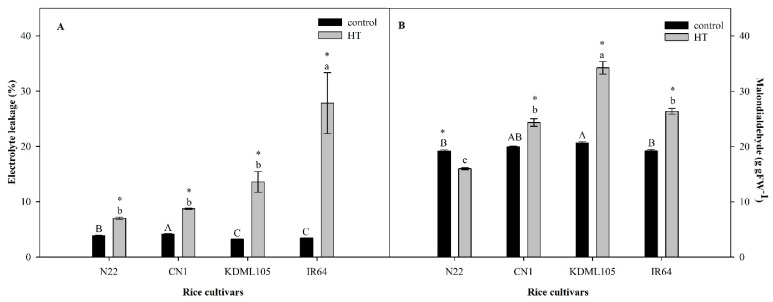
Effect of HT on electrolyte leakage (**A**) and lipid peroxidation (**B**) in four rice cultivars (N22, CN1, KDML105 and IR64). Each bar represents mean ± SE (*n* = 3–4). Asterisk (*) indicates significant difference between control and HT in each rice cultivar at *p* ≤ 0.05 by *t* test. Different capital letters indicate significant difference of control group in different rice cultivars at *p* ≤ 0.05 by ANOVA and DMRT, and small letters indicate significant difference of HT in different rice cultivars at *p* ≤ 0.05 by ANOVA and DMRT.

**Figure 3 plants-11-01454-f003:**
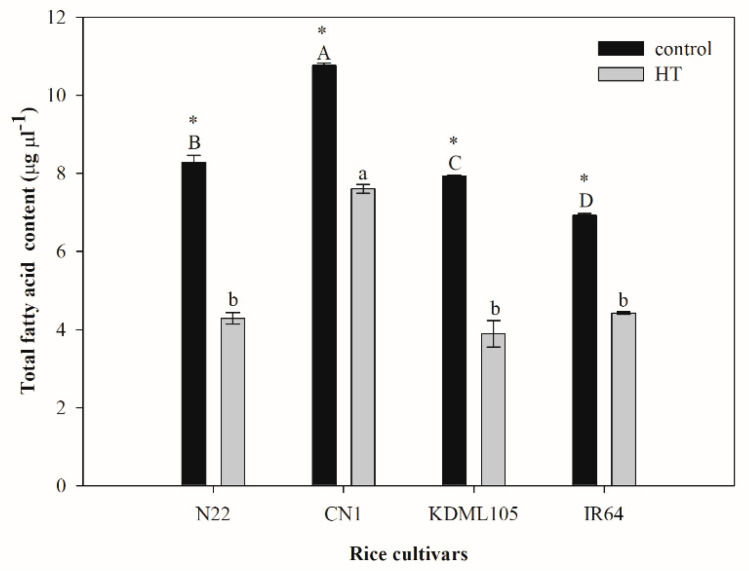
Effect of HT on total fatty acids in four rice cultivars (N22, CN1, KDML105 and IR64). Each bar represents mean ± SE (*n* = 3–4). Asterisk (*) indicates significant difference between control and HT in each rice cultivar at *p* ≤ 0.05 by *t* test. Different capital letters indicate significant difference of control group in different rice cultivars at *p* ≤ 0.05 by ANOVA and DMRT, and small letters indicate significant difference of HT in different rice cultivars at *p* ≤ 0.05 by ANOVA and DMRT.

**Figure 4 plants-11-01454-f004:**
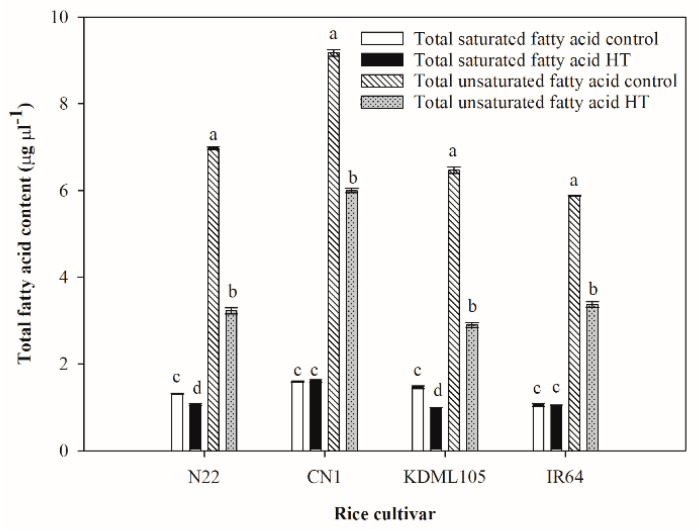
Effect of HT on total saturated and total unsaturated fatty acids in four rice cultivars (N22, CN1, KDML105 and IR64). Each bar represents mean ± SE (*n* = 3–4), and different small letters indicate significant difference in each rice cultivar at *p* ≤ 0.05 by ANOVA and DMRT.

**Figure 5 plants-11-01454-f005:**
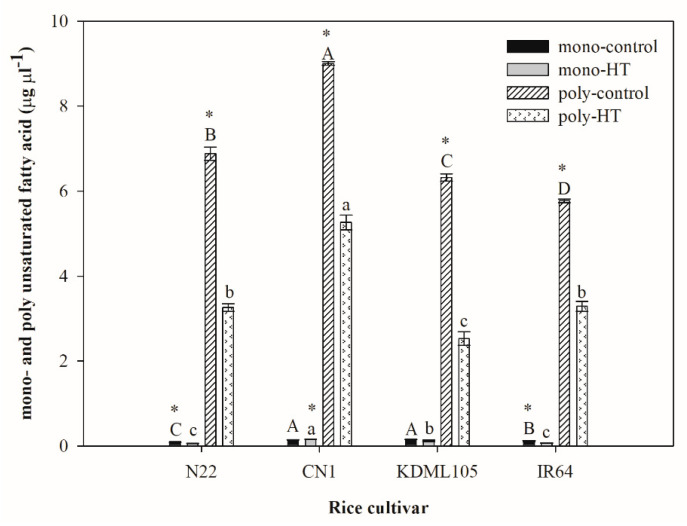
Effect of HT on mono- and polyunsaturated fatty acids in four rice cultivars (N22, CN1, KDML105 and IR64). Each bar represents mean ± SE (*n* = 3–4). Asterisk (*) indicates significant difference between control and HT in mono- or poly unsaturated fatty acid and in each rice cultivar at *p* ≤ 0.05 by *t* test. Different capital letters indicate significant difference of control group in different rice cultivars at *p* ≤ 0.05 by ANOVA and DMRT, and small letters indicate significant difference of HT in different rice cultivars at *p* ≤ 0.05 by ANOVA and DMRT.

**Table 1 plants-11-01454-t001:** Saturated fatty acids in the control and HT-stressed leaves of four rice cultivars.

Saturated Fatty Acids (µg µL^−1^)	N22		CN1		KDML105		IR64	
	Control	HT	Control	HT	Control	HT	Control	HT
Capric acid (C10:0)	0.000f	0.000f	0.006f	0.008d *	0.006f	0.010c *	0.000e	0.000g
Lauric acid (C12:0)	0.076c *	0.047d	0.093c *	0.087c	0.080e *	0.050c	0.050d	0.052e
Myristic acid (C14:0)	0.038d *	0.023e	0.028e	0.030d *	0.021f	0.031c	0.015e	0.018f
Pentadecanoic acid (C15:0)	0.017e *	0.012ef	0.025e	0.029d *	0.019f	0.021c	0.015e	0.015f
Palmitic acid (C16:0)	0.768a *	0.498a	1.048a *	0.951a	0.762a *	0.550a	0.655a *	0.548a
Stearic acid (C18:0)	0.161b *	0.119b	0.185b *	0.150b	0.156bc	0.138b	0.119b *	0.089d
Arachidic acid (C20:0)	0.088c *	0.128b	0.057d	0.104c *	0.131cd *	0.057c	0.060cd	0.109c *
Behenic acid (C22:0)	0.080c *	0.133b	0.055d	0.110c *	0.109de *	0.035c	0.066cd	0.116b *
Lignoceric acid (24:0)	0.085c	0.104c	0.097c	0.138b *	0.178b *	0.103b	0.071c	0.108c *
Total	1.313C *	1.064b	1.594A	1.606a	1.462B *	0.995b	1.050D	1.055b

Asterisk (*) indicates significant difference between control and HT in each rice cultivar at *p* ≤ 0.05 by *t* test. Different capital letters indicate significant difference of control group in different rice cultivars at *p* ≤ 0.05 by ANOVA and DMRT, and small letters indicate significant difference of HT in different rice cultivars at *p* ≤ 0.05 by ANOVA and DMRT. Superscript small letters indicate significant difference in the same column at *p* ≤ 0.05 by ANOVA and DMRT.

**Table 2 plants-11-01454-t002:** Unsaturated fatty acids in the control and HT-stressed leaves of four rice cultivars.

Unsaturated Fatty Acids (µg µL^−1^)	N22		CN1		KDML105		IR64	
	Control	HT	Control	HT	Control	HT	Control	HT
Linoleic acid (C18:2)	0.709b *	0.298b	1.012b *	0.625b	0.668b	0.567b	0.622b *	0.381b
γ-Linolenic acid (C18:3n6)	0.027c *	0.010b	0.033d *	0.022c	0.023d *	0.012c	0.018d *	0.012c
Linolenic acid (C18:3)	6.137a *	2.855a	7.981a *	5.174a	5.613a *	2.199a	5.099a	2.910a
cis-11,14-Eicosadienoic acid (C20:2)	0.003c *	0.000c	0.005d *	0.000c	0.005d *	0.000c	0.003d *	0.000c
cis-5,8,11,14,17-Eicosapentaenoic acid (C20:5)	0.008c *	0.008c	0.002d	0.013c *	0.012d *	0.008c	0.006d	0.010c
Palmitoleic acid (C16:1)	0.004c	0.006c *	0.000d	0.085c *	0.000d	0.046c *	0.005d	0.006c *
Oleic acid (C18:1)	0.072c *	0.030c	0.132c *	0.069c	0.133c *	0.058c	0.109c *	0.046c
Erucic acid (C22:1)	0.014c	0.016c	0.009d	0.013c *	0.017d *	0.008c	0.014d *	0.004c
Total	6.974B *	3.224bc	9.174A *	6.002a	6.471C *	2.897c	5.875D *	3.370b

Asterisk (*) indicates significant difference between control and HT in each rice cultivar at *p* ≤ 0.05 by *t* test. Different capital letters indicate significant difference of control group in different rice cultivars at *p* ≤ 0.05 by ANOVA and DMRT, and small letters indicate significant difference of HT in different rice cultivars at *p* ≤ 0.05 by ANOVA and DMRT. Superscript small letters in the same column indicate significant difference between different unsaturated fatty acid at *p* < 0.05 by ANOVA and DMRT.

**Table 3 plants-11-01454-t003:** Ratio of saturated and unsaturated fatty acids in different seedling rice cultivars (N22, CN1, KDML105 and IR64) after exposure to high temperature at 42 °C for 7 days.

Rice Cultivars	Total Sat		Total Unsat		Sat:Unsat		% Increase in Sat:Unsat after HT
	Control	HT	Control	HT	Control	HT	
N22	1.312	1.064	6.974	3.224	0.188b	0.330a *	+75%a
CN1	1.595	1.606	9.174	6.002	0.174b	0.268b *	+54%b
KDML105	1.462	0.995	6.471	2.897	0.226a	0.346a *	+53%b
IR64	1.050	1.055	5.875	3.370	0.179b	0.313ab *	+75%a

Asterisk (*) indicates significant different between control and HT in each rice cultivar at *p* ≤ 0.05 by independent-samples *t* test. Different small letter in the same column indicate significant difference between different rice cultivars at *p* ≤ 0.05 by ANOVA and DMRT.

**Table 4 plants-11-01454-t004:** Temperature control data and relative humidity in the growth chamber (VRV.Corp., Ltd., Bangkok, Thailand).

No.	Time (h)	Temperature (°C)	Relative Humidity (%)
1	00:00–03:00	27	66
2	03:00–07:00	26	70
3	07:00–09:00	29	61
4	09:00–11:00	36	49
5	11:00–15:00	42	42
6	15:00–17:00	40	40
7	17:00–18:00	37	37
8	18:00–21:00	33	33
9	21:00–00:00	27	32

Temperature and relative humidity were obtained from Agronomy station, Faculty of Agriculture, Khon Kaen University during March—May 2016 and 2017.

**Table 5 plants-11-01454-t005:** Temperature, humidity and light intensity control data in the growth chamber (VRV.Corp., Ltd., Thailand).

No.	Time (h)	* Light Intensity (µmol m^−2^ s^−1^)
1	06:00–07:00	70
2	07:00–08:00	115
3	08:00–09:00	200
4	09:00–10:00	265
5	10:00–11:00	340
6	11:00–13:00	390
7	13:00–14:00	340
8	14:00–15:00	265
9	15:00–16:00	200
10	16:00–17:00	115
11	17:00–18:00	70
12	18:00–06:00	0

Light intensity was obtained from Agronomy station, Faculty of Agriculture, Khon Kaen University during March—May 2016 and 2017. * Light intensity was measured by light incident on the rice leaves at 30 cm from the light source bulb.

## Data Availability

Not applicable.

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
