# Peer review of "High Temperature Alters Leaf Lipid Membrane Composition Associated with Photochemistry of PSII and Membrane Thermostability in Rice Seedlings"

_plants, 2022, doi:10.3390/plants11111454_

Round 1

Reviewer 1 Report

The authors studied four rice cultivars with different heat tolerance exposed to high temperature. They investigated the physiological parameters and leaf fatty acid compositions in response to high temperature. They found that increased long-chain fatty acids may lead to cuticular wax synthesis which provides protective roles for heat tolerance. Overall, this is an interesting topic that fatty acids can act as intermediates for plant heat tolerance.

Here are my questions.

  1. The authors performed the heat treatment by growing seedlings at 42℃/27℃ (day/night) for 7 days. Since heat stress can induce lipid remodeling in plants, the authors should analyze or discuss the fatty acid dynamics during heat processing of seedlings. This is also because the different responsive patterns between prolonged warming and heat shock in plants (Wang et al, BMC Plant Biology 2020, 20, 86).
  2. After high temperature treatment, were there any phenotypic or morphological changes/differences between four cultivars and control, especially in heat sensitive cultivars? If there were some suffer symptoms, the authors should provide the photos of seedlings.
  3. “Dramatic increases in long-chain fatty acids may lead to cuticular wax synthesis which provides protective roles for heat tolerance.” Have the authors measured the thickness or contents of cuticular wax in the leaves? Such as N22 (heat tolerant) vs IR64 (heat sensitive), HT treatment vs control.

Minors

Line 22-23, this sentence is grammatically wrong.

What sizes of four cultivars were grown for 28 days in a greenhouse? Were they in similar sizes?

Fifty seedlings were grown in a pot. I wonder whether these seedlings could influence each other within the population. Some studies have showed the plants can warning each other when encounter the stress.

Author Response

Dear Reviewer 1

Thank you so much for valuable suggestions of our MS and we are pleased to submit the revised version of plants-1736864 entitled “High temperature alters leaf lipid membrane composition associated with photochemistry of PSII and membrane thermostability in rice seedlings” for intended publication in Plants for your kind consideration. We appreciated the constructive criticism of the reviewers. As comments of reviewers, we tried to revised carefully following reviewer’s suggestions in track changes for adding more information in the text. Our responses were given point by point as outlined below.

The Responses to reviewers ’s comments

Reviewer 1’s comments

  1. The authors performed the heat treatment by growing seedlings at 42℃/27℃ (day/night) for 7 days. Since heat stress can induce lipid remodeling in plants, the authors should analyze or discuss the fatty acid dynamics during heat processing of seedlings. This is also because the different responsive patterns between prolonged warming and heat shock in plants (Wang et al, BMC Plant Biology 2020, 20, 86).

Thank you so much for your constructive suggestions. We have added more discussion about fatty acid dynamics citing the reference that you suggested, and two more recent reference (Zoong Lwe et al. 2020, and 2021).

  1. After high temperature treatment, were there any phenotypic or morphological changes/differences between four cultivars and control, especially in heat sensitive cultivars? If there were some suffer symptoms, the authors should provide the photos of seedlings.

Yes, there were some morphological changes of leaves of heat-treated rice plants compared to those of the controls. For example, we found that after heat treatment rice cv. IR64 showed more severe symptoms of leaf drooping, leaf burning and drying compared to other cultivars. We added the description of morphological changes in the revised version. However, the photographs we took were not of good enough quality for publication, so we decide not to present the figures.

  1. “Dramatic increases in long-chain fatty acids may lead to cuticular wax synthesis which provides protective roles for heat tolerance.” Have the authors measured the thickness or contents of cuticular wax in the leaves? Such as N22 (heat tolerant) vs IR64 (heat sensitive), HT treatment vs control.

Thank you so much for your comments. Unfortunately, we did not measure the thickness or contents of cuticular wax in the leaves. However, the fact that we analysed fatty acids from the whole leaf and found the increase in many long-chain fatty acids not previously reported in publications based on fatty acid analysis from thylakoid membrane, is considered a new finding. According to your suggestions, we planned to investigate in more details the changes in cuticle thickness and cuticular wax contents under heat stress. 

  1. Line 22-23, this sentence is grammatically wrong.

This sentence has been modified following the suggestion of Reviewer #2 as follows:- “Thus, the reduction in unsaturated fatty acid composition of the thylakoid membrane and dramatic increases in long-chain fatty acids may lead to high photosynthetic performance and an enhanced synthesis of cuticular wax which further provided additional protective roles for heat tolerance ability in rice.”

  1. What sizes of four cultivars were grown for 28 days in a greenhouse? Were they in similar sizes?

The sizes of the four cultivars were different at the age of 28 days because they were the constitutive characteristics of each cultivar. The sizes of four rice cultivars (from larger to smaller) were in the order: KDML105 > CN1 > N22 > IR64. Stress tolerance ability of rice was not related to plant size, in this case, KDML105 was more heat sensitive than CN1 and N22 despite its larger size.

  1. Fifty seedlings were grown in a pot. I wonder whether these seedlings could influence each other within the population. Some studies have showed the plants can warning each other when encounter the stress.

The reason for using fifty seedlings per pot was because a large amount of leaves (50 g) was needed for fatty acid analysis. Your suggestions about the interaction and warning among plants are intuitive and highly appreciated, and we will take this into consideration for improving our future pot experiments.

Best regards,

Anoma Dongsansuk

Reviewer 2 Report

Although the paper might have some interest to the agrofood sector dealing with rice, there are several points that the authors must consider prior to publication. In fact, in the present form the paper is mostly technical than a research paper (it deals with "the effect of this .... on that"), yet has some interesting data that should be discussed in a more deep perspective. In the abstract, the aims must be objective and focusing on the main hypothesis under investigation (and not considering the aims reported in the last sentence of the introduction). Besides, a general conclusion must be also provided in the abstract, which might give a further perspective about the relevance of the study. The introduction must consider updated works published by key researchers working on the scientific field, but there are in the introduction and along the paper a prevalence of references of relatively old papers. So, the introduction must be updated. The materials and methods section can be consider ok, but some information is still missing relatively to the growth condictions, namely the applied PAR (Photosynthetically Active Radiance), GPS coordinates of the greenhouse (since it appears that natural light conditions were used (?). Results seem robust and can be accept in the present form. Discussion is merely a superficial analysis - the authors must be a closer relation with the stress and the parametric metabolisms. The conclusion must not be a simple repetition of data previously reported but a deeper analysis about the valorization / application of the study. Besides, the English style is poor and must be improved.

Author Response

Dear Reviewer 2

Thank you so much for valuable suggestions of our MS and we are pleased to submit the revised version of plants-1736864 entitled “High temperature alters leaf lipid membrane composition associated with photochemistry of PSII and membrane thermostability in rice seedlings” for intended publication in Plants for your kind consideration. We appreciated the constructive criticism of the reviewers. As comments of reviewers, we tried to revised carefully following reviewer’s suggestions in track changes for adding more information in the text. Our responses were given point by point as outlined below.

The Responses to reviewers ’s comments

Reviewer 2’s comments

  1. Although the paper might have some interest to the agrofood sector dealing with rice, there are several points that the authors must consider prior to publication. In fact, in the present form the paper is mostly technical than a research paper (it deals with "the effect of this .... on that"), yet has some interesting data that should be discussed in a more deep perspective.

We added more discussion regarding the mechanisms of reduction in PSII efficiency, and fatty acid dynamics under heat stress.

  1. In the abstract, the aims must be objective and focusing on the main hypothesis under investigation (and not considering the aims reported in the last sentence of the introduction).

In the abstract, the sentence specifying the aims has been improved as follows:- “This study aimed at characterizing the HT response in terms of PSII efficiency and membrane stability, and to identify leaf fatty acid changes that may be associated with HT tolerance or sensitivity of rice genotypes.”

  1. Besides, a general conclusion must be also provided in the abstract, which might give a further perspective about the relevance of the study.

We revised the conclusion of abstract as “Thus, the reduction in unsaturated fatty acid composition of the thylakoid membrane and dramatic increases in long-chain fatty acids may lead to high photosynthetic performance and an enhanced synthesis of cuticular wax which further provided additional protective roles for heat tolerance ability in rice.”

  1. The introduction must consider updated works published by key researchers working on the scientific field, but there are in the introduction and along the paper a prevalence of references of relatively old papers. So, the introduction must be updated.

We added 3 more recent references in the introduction part regarding the remodeling fatty acid composition (Zoong Lwe et al., 2020 and 2021; and Narayanan et al., 2020).

  1. The materials and methods section can be considered ok, but some information is still missing relatively to the growth condictions, namely the applied PAR (Photosynthetically Active Radiance), GPS coordinates of the greenhouse (since it appears that natural light conditions were used (?).

We added PAR and GPS data in the materials and methods.

  1. Results seem robust and can be accept in the present form.

Thank you very much for your compliments.

  1. Discussion is merely a superficial analysis - the authors must be a closer relation with the stress and the parametric metabolisms.

We added more discussion regarding the mechanisms of reduction in PSII efficiency, and changes in lipid compositions under heat stress.

  1. The conclusion must not be a simple repetition of data previously reported but a deeper analysis about the valorization / application of the study.

We added the conclusion for the application of this study as “From this study, changes in leaf fatty acid compositions can potentially be used as indicators for heat tolerance in rice.”

  1. Besides, the English style is poor and must be improved.

Our MS was edited English language by native speaker. Please find attached the proofreading certificate.

Round 2

Reviewer 1 Report

All my questions and concerns have been addressed.

Reviewer 2 Report

According to my perspective the new version of the paper complies with the high standard of the journal and can be published